# Women with Autoimmune Thyroiditis Taking Levothyroxine During Pregnancy: Is Iodine Supplementation Needed?

**DOI:** 10.3390/nu17030542

**Published:** 2025-01-31

**Authors:** Simona Censi, Giulia Messina, Emma Feligiotti, Cristina Clausi, Ilaria Piva, Daniela Basso, Isabella Merante Boschin, Loris Bertazza, Fiammetta Battheu, Susi Barollo, Marta Camilot, Caterina Mian

**Affiliations:** 1Endocrinology Unit, Department of Medicine (DIMED), University of Padua, 35121 Padua, Italy; simona.censi@unipd.it (S.C.); giulia.messina.1@studenti.unipd.it (G.M.); emmafelig@gmail.com (E.F.); cristina.clausi@aopd.veneto.it (C.C.); piva.ilaria.92@gmail.com (I.P.); loris.bertazza@unipd.it (L.B.); fiammetta.battheu@aopd.veneto.it (F.B.); susi.barollo@unipd.it (S.B.); 2Laboratory Medicine, Department of Medicine (DIMED), University of Padua, 35121 Padua, Italy; daniela.basso@unipd.it; 3Department of Medical and Surgical Sciences (DiSCOG), University of Padua, 35100 Padua, Italy; isabella.meranteboschin@unipd.it; 4Department of Paediatrics, Regional Centre for Newborn Screening, Diagnosis and Treatment of Inherited Metabolic Diseases and Congenital Endocrine Diseases, Azienda Ospedaliera Universitaria Integrata, 37134 Verona, Italy; marta.camilot@aovr.veneto.it

**Keywords:** iodine, pregnancy, diet, supplements

## Abstract

Background: Iodine is fundamental for the synthesis of thyroid hormones, which play a central role in foetal neurological development. The need for an iodine-containing supplement (ICS) in L-T4-treated women during pregnancy is still a subject of debate. Aim of the Study: The aim of the study is to investigate the iodine status in women with autoimmune thyroiditis (AT) who have or have not been treated with L-T4. Methods: This was a cross-sectional, observational study involving pregnant women with AT, treated with/without L-T4. Upon enrolment, women provided a urine sample (to measure the urinary iodine concentration (UIC), which was normalised to urinary creatinine values (UI/Creat)), and completed a questionnaire. TSH, FT4, and neonatal TSH were also obtained. Results: Among women taking an ICS, 74.1% had a UI/Creat level ≥ 150 μg/g, compared with only 46.2% of those not taking an ICS (*p* = 0.03). Among L-T4 users only, a UI/Creat level ≥ 150 μg/g was more frequent in ICS users than in non-ICS users (72.8% vs. 41.7%) (*p* = 0.03). In the multivariate analysis, ICS use was the only independent variable for UI/Creat ≥ 150 μg/g (OR: 3.4; CI: 1.1–10.9) (*p* = 0.04). There was a tendency towards higher UI/Creat levels as the L-T4 (µg/Kg) dosage increased, although no correlation was found. Newborns of women taking an ICS were found to have elevated neonatal TSH (2.8 mIU/L vs. 1.7 mIU/L) (*p* = 0.04). All newborns with a TSH >5 mUI/L were those of women taking supplements. Conclusions: Women with AT taking L-T4 still need iodine supplementation, although the amount should be regulated on the basis of their L-T4 dosage.

## 1. Introduction

Iodine is an essential micronutrient for normal thyroid hormone biosynthesis and is mainly obtained from the diet. In pregnancy, an approximately 50% increase in iodine is required to achieve a dietary intake of around 250 µg/day. This increase is due to increased glomerular filtration and renal iodine clearance, the rise in thyroxine-binding globulin concentrations, the effects of hCG on the maternal thyroid, the role of the placenta in deiodination of iodothyronines, and iodine transplacental transfer to the foetus, particularly in later gestation [1]. Iodine deficiency and, consequently, inadequate iodine availability for the foetus, is responsible for induced iodine deficiency disorders (IDDs, and carries an increased risk of maternal goitre, foetal goitre, miscarriage, growth retardation, and brain damage [2,3].

Therefore, it is very important to closely monitor iodine status during pregnancy and to ensure an adequate daily iodine intake in order to reach the recommended requirement of 250 μg/L. Urinary iodine concentration (UIC) is an indirect marker of iodine availability and is a widely used indicator of population iodine status, given that 90% of daily iodine intake is excreted in the urine. According to World Health Organization (WHO) guidelines, median urinary iodine values ≥150 µg/L represent an inadequate iodine intake for pregnant women [4,5]. While both the American and European Thyroid Associations (ATA and ETA) recommend iodine supplementation (150 µg/day) in pregnancy, the WHO does not consider it necessary for women living in countries running effective and sustained iodised salt programmes [5,6]. However, no consensus has been reached yet on whether supplementation should also be recommended in pregnant women with Hashimoto’s thyroiditis who are receiving levothyroxine (L-T4) therapy. In particular, WHO guidelines do not distinguish between pregnant women with and without Hashimoto’s thyroiditis, while ETA guidelines state that the effects of supplementation in hypothyroid women—whether undergoing L-T4 therapy or not—need to be further investigated [7]. ATA guidelines, instead, state that there is no need for women undergoing L-T4 therapy to take a supplement, although this is a weak recommendation with a low level of evidence. It is therefore still not clear when and how pregnant women with Hashimoto’s thyroiditis should take an iodine supplement.

The principal aim of our study was to evaluate the iodine status in a sample of pregnant women residing in the Veneto region of Italy and suffering from autoimmune thyroiditis in relation to dietary habits (the use of iodised salt (IS)), the use of an iodine-containing supplement (ICS), and LT-4 therapy.

## 2. Materials and Methods

We conducted a cross-sectional, observational study. Women were consecutively recruited at the time of their pregnancy endocrinological consultation, from February 2021 to December 2023. Inclusion criteria were as follows: (i) resident in the Veneto region; (ii) pregnant with a single pregnancy; (iii) ≥18 years of age; (iv) having autoimmune thyroiditis; (v) giving informed consent. Exclusion criteria were as follows: (i) refusing informed consent; (ii) presence of a language barrier; (iii) hyperthyroidism or hypothyroidism of a non-autoimmune nature and previous thyroidectomy or ablative therapy with iodine-131. Upon enrolment, the subjects provided an early-morning spot urine sample to assay urinary iodine and creatinine concentrations (the UI/Creat ratio was used to measure iodine status). TSH and FT4 were obtained from electronic databases, as they are part of the routine management of thyroid autoimmune disease. Thyroid ultrasound was performed with an EUB 7500 HV machine (Hitachi Medical Corporation 4-14-1, Soto-Kanda, Chiyoda-ku, Tokyo, Japan) with a 7.5 mHz linear electronic transducer to assess anteroposterior diameters (APd) and echogenicity.

The women were also administered a questionnaire (Appendix A of Supplemental Material) to collect personal and anthropometric data; pregnancy data; lifestyle information; use of iodised salt (IS); use of an ICS and the brand name (to gather information on the quantitative iodine content); and a question regarding knowledge of the iodine prophylaxis programme.

The newborns’ TSH levels were obtained from the congenital hypothyroidism screening programme. The following information on newborns was gathered from electronic databases: gestational age, sex, and weight and height at birth. The study was performed in accordance with the guidelines of the Helsinki Declaration and was approved by the Local Ethical Committee (Padua General Hospital, Comitato Etico per la Sperimentazione Clinica (CESC), code number: 4603/AO/18b, 11 December 2019). All patients gave their written informed consent.

### 2.1. Laboratory Assays

The early-morning, non-fasting urine samples were collected, divided into aliquots, and refrigerated at −20 °C until assay. UIC was measured in duplicate using the ceric arsenious acid reaction in a Technicon Auto-Analyzer (Brain Luebbe GmbH, Norderstedt, Germany) [8]). Between-run accuracy, assessed by repeatedly measuring two levels of in-house quality control material (95 and 300 μg/L), was from 1.7% to 2.8%. At the start of each run, a standard curve was also verified with serial dilutions of a synthetic tester. Commercial kits (Roche, Rotkreuz, Switzerland) were used to measure serum TSH, FT3, and FT4 levels, with the following reference ranges: TSH: 0.2–4 mIU/L; FT3: 3.90–6.80 pmol/L; FT4: 9.00–22.00 pmol/L. Maternal serum was collected to assess thyroid function at the same time as urinary iodine concentrations were evaluated. An enzymatic method (Roche Cobas 8000 Modular Analyzer, Indianapolis, IN, USA) was used to measure urinary creatinine, with a reference range of 0.1–54 mmol/L. Finally, UI/Creat was measured in μg/g.

### 2.2. Statistical Analysis

The Kolmogorov–Smirnov test was used to determine the normal distribution of each variable. As the quantitative variables were non-normally distributed, they were reported as medians and interquartile ranges (IQR). Mann–Whitney or Kruskal–Wallis tests were used, as appropriate, to explore the relationships between the quantitative and categorical variables. The χ^2^ test was used for categorical variables. The Pearsons correlation test was performed after the logarithmic transformation of the variables to correlate UI/Creat with the daily dosage (μg/Kg) of L-T4. The factors influencing iodine status (< or ≥150/250/500 μg/g) were identified using multivariate analysis with stepwise logistic regression analysis. The factors influencing TSH were explored using a multiple linear regression analysis model. The size of the effect on the continuous variables was calculated with Cohen’s d index. Since multiple comparison could be a limitation of the present study, we applied a false discovery rate test (FDR) for multiple testing (median UI/Creat and UI/Creat ≥ 150 µg/g) to ensure that our results were not obtained from multiple tests. A *p*-value of <0.05 was considered statistically significant. The statistical analyses were performed in the R software version 2.7.2.

## 3. Results

### 3.1. Iodine Status

We enrolled 125 women with a median age of 35.0 years (IQR: 31.6–38.3 years) and at a median gestational age of 24 weeks (IQR: 20.0–29.5 weeks). The subjects’ characteristics are reported in Table 1. Median UI/Creat was 263.0 μg/g (IQR: 128–471.0 μg/g), and 89/125 women (71.2%) had a UI/Creat ≥ 150 μg/g. More specifically, 36/125 (28.8%) had a UI/Creat < 150 μg/g, 20/125 (16.0%) had a UI/Creat ≥ 150 μg/g and <250 μg/g, 69/125 (55.2%) had a UI/Creat ≥ 250 μg/g, and 31/125 (24.8%) had a UI/Creat ≥ 500 μg/g.

### 3.2. Iodine Status and Knowledge of Iodine Prophylaxis Campaign

Only 12/122 (9.8%) of the interviewed women knew about the iodine prophylaxis campaign. There were no differences in iodine status in relation to this knowledge (Table 2).

### 3.3. Iodine Status and Levothyroxine Therapy

One hundred and fifteen women were on L-T4 therapy (92%). Patients taking L-T4 had higher UI/Creat levels than those not taking it (267.0 μg/g (IQR: 117.0–498.5 μg/g) versus 175.0 μg/g (IQR: 164.5–378.1 μg/g)), but the difference was not statistically significant (*p* = 0.6). There were higher percentages of L-T4 users than non-users among women with a more-than-adequate iodine status (94.2% versus 5.8%) and among those with excessive UI/Creat (96.8% versus 3.2%), although the differences were not significant, possibly because of the very low number of non-L-T4 users (*p* = 0.3) (Table 3 and Table 4). We observed a tendency towards higher UI/Creat levels as the L-T4 dosage (µg/Kg/day) increased, although no correlation was found (correlation coefficient: 0.174, (95% CI: −0.009–0.346), *p* = 0.06 (Figure 1).

### 3.4. Iodine Status, Dietary Habits, and Supplements

An ICS was being taken by 112/125 (89.6%) of the women. The median iodine content of the iodine supplement was 220 μg (IQR: 175–220 μg). All the women not on L-T4 therapy were taking a supplement. There was a higher frequency of UI/Creat ≥ 150 μg/g in ICS users than in non-users (74.1% (83/112) versus 46.2% (6/13)) (*p* = 0.03), but, while the median UI/Creat was higher in the former than in the latter, the difference was not statistically significant (Table 2). Among women with a UI/Creat ≥ 250 μg/g, the vast majority (65/69 (94.2%)) were taking an ICS, although again without statistical significance (*p* = 0.06) (Table 3).

Looking only at L-T4 users, an adequate iodine status (≥150 μg/g) was more frequent among ICS users than among non-users: 75/103 (72.8%) versus 5/12 (41.7%) (*p* = 0.03). Similarly, there was a higher frequency of UI/Creat ≥250 μg/g among ICS users than among non-users: 61/103 (59.2%) versus 4/12 (33.3%), but without statistical significance (*p* = 0.09).

Forty-five women out of one hundred and twenty-two (36.9%) regularly consumed cows’ milk (at least 1 cup/day). Iodine status was higher among regular cows’ milk consumers (median UI/Creat 342.0 μg/g (IQR: 195.0–603.0 μg/g)) than among non-consumers (239.0 μg/g (IQR: 115.0–436.0 μg/g)) (*p* = 0.03, Cohen’s d index: 0.306 (95% CI: 0.675–0.065)). There was also a higher frequency of UIC/Creat ≥ 150 μg/g among the former than the latter (37/45 (82.2%) versus 50/77 (64.9%)) (*p* = 0.04) (Table 2).

A multivariate logistic regression analysis model (that included the dietary covariates having a relationship that was significant or at the limits of significance at univariate analysis, i.e., ICS and regular cows’ milk consumption) revealed that only ICS was an independent predictor of adequate iodine status with an odds ratio (OR) of 3.4 (95% CI: 1.0–10.9) (*p* = 0.04). Taking a more-than-adequate iodine status (UI/Creat ≥ 250 μg/g) as the outcome, a logistic regression analysis model, which included as covariates those factors found to be more closely related with the outcome (as shown in Table 3), showed there to be no independent factors of a more-than-adequate iodine status.

With regard to the 31 women with excessive UI/Creat, no significant factors were identified as being associated with the excess, neither at the univariate (Table 4) nor at the multivariate logistic regression analyses (with those factors found to be more closely related with the outcome as covariates, as shown in Table 4).

### 3.5. Iodine Status and Thyroid Maternal Function

Data on maternal thyroid function were available for 117 cases. Median TSH values were 2.1 mU/L (IQR: 1.4–2.8 mU/L). No correlation was found between TSH and UI/Creat values. As expected, TSH correlated with the daily dosage (μg/Kg/due) of L-T4 (*p* = 0.01, r = 0.22). Median FT4 was 12.0 pmol/L (IQR: 10.4–13.5 pmol/L) and did not correlate with UI/Creat. Women not using an ICS showed a tendency towards higher TSH values compared with ICS users (2.7 mUI/L (IQR: 1.8–3.3) versus 1.9 mUI/L (IQR: 1.4–2.7)) (*p* = 0.08). However, the multiple linear regression analysis model (with L-T4, use of ICS and UI/Creat values included as covariates) revealed only L-T4 to be an independent parameter of TSH values (β = 1.288, *p* = 0.01). In 35/125 (28.0%) women, the autoimmune thyroiditis was diagnosed during their pregnancy endocrinological consultations, while in 72% of cases, it was a previous diagnosis. In all cases, the diagnosis was confirmed by ultrasound (US) echogenicity. The median of the sum of the anteroposterior diameters (right and left) was 28 mm (IQR: 24–32 mm). A negative but non-significant correlation was found between the sum of the diameters and UI/Creat (*p* = 0.6, r = −0.5), with a trend towards higher UI/Creat values in women with a known history of thyroiditis (and thus a long-term AT background) than in women in whom it was discovered during pregnancy: 278.5 μg/g (IQR: 136.0–506.0 μg/g) versus 258.0 μg/g (IQR: 107.5–446.5 μg/g) (*p* = 0.4). Women with a known history of AT had smaller thyroids than women with an AT diagnosed during pregnancy, the sum of the diameters being 28mm (IQR: 23.0–31.8 mm) and 30.0 (IQR: 27.0–32.0), respectively, at the limits of significance (*p* = 0.07).

### 3.6. Iodine Status, Neonatal TSH, and Pregnancy Outcomes

The median gestational period at birth was 39 weeks (IQR: 38–40 weeks). Nine out of one hundred and thirteen (7.9%) pre-term deliveries were documented. Women with a pre-term delivery had lower median UI/Creat values compared with their counterparts, at 182.0 μg/g (IQR: 109.3–484.0 μg/g) versus 267.0 μg/g (IQR: 128.3–519.5 μg/g), although the difference was not significant (*p* = 0.6). The median birth weight was 3300 g (IQR: 3028.3–3630.0 g), median head circumference was 34 cm (IQR: 33.0–35.5 cm), and median birth height was 49.0 cm (IQR: 48.0–51.0 cm). No correlations were obtained between these parameters and maternal UI/Creat. Neonatal TSH was available for 110 newborns. The median neonatal TSH was 2.8 mUI/L (IQR: 1.9–3.7 mUI/L). In 8 out of 110 (7.2%), TSH was >5 mUI/L. Higher neonatal TSH values were documented in newborns of women taking an ICS than in those not taking an ICS, although they were within normal ranges: 2.8 mUI/L (IQR: 1.9–3.7 mUI/L) and 1.7 mUI/L (IQR: 1.1–2.9), respectively (*p* = 0.04). Moreover, all eight cases of neonatal TSH > 5 mUI/L were in newborns of mothers taking an ICS. No differences were found based on the use of L-T4.

## 4. Discussion

Iodine deficiency during pregnancy may result in inadequate thyroid hormone synthesis in both the mother and the foetus, with consequences for maternal and foetal outcomes [9,10]. On the other hand, an excess of iodine may also be harmful to the foetus, since it can lead to the persistent blocking of thyroid hormone synthesis, causing hypothyroidism and goitre [11]. Furthermore, the foetal thyroid may be more susceptible to fluctuations in iodine exposure as the regulatory mechanisms are not yet fully mature [4,12]. Many guidelines specifically recommend iodine supplementation during pregnancy, although they give no explicit recommendations for women on L-T4 therapy [5,6,7]. It is well known that L-T4 therapy itself provides a non-negligible quantity of iodine [13] that adds to the amount obtained from the diet and from iodine supplements where taken. However, few data on this issue are available in the literature. A single Polish study involving 92 pregnant women on L-T4 therapy, supplemented or not with 150 μg/day, found a higher median UIC in the supplemented women, the only group to attain iodine sufficiency (median UIC ≥ 150 μg). Iodine supplementation had no impact on maternal thyroid function. Median values of neonatal TSH were the same for the two groups, although all five cases of neonatal TSH > 5 mUI/L were in newborns of supplemented women [14]. We obtained similar results. In women taking L-T4, there was a trend towards a higher median UI/Creat and a greater frequency of a UI/Creat ≥ 150 μg among ICS users compared with non-ICS users. It is interesting that, despite the iodine obtained from L-T4 therapy, treated patients still needed an ICS to reach iodine sufficiency during pregnancy. On the other hand, the vast majority of patients with a more-than-adequate or excessive iodine status were undergoing L-T4 therapy, suggesting that it plays a role in potential iodine excess. In this regard, we found a positive correlation at the limits of significance between UI/Creat and the daily L-T4 dosage. It is important to mention that most of our subjects (almost 90%) were taking an ICS during their pregnancy, regardless of whether or not they had an autoimmune thyroid background. The majority of the supplements on the market in Italy contain iodine among their different components, and, in fact, 55% of the women in our study had UIC values above 250 μg/g.

All these data put together suggest that iodine supplementation in pregnant women on L-T4 is still necessary, at least in areas of mild-to-moderate iodine deficiency, such as our region, but it should be adapted to the dose of L-T4 taken by each woman. Unfortunately, we were unable to determine the optimal dose of iodine supplementation due to both the small size of our sample and the fact that commercially available pregnancy iodine supplements have fixed iodine contents (i.e., 150/175/200/220 μg), which precluded a potential statistical correlation.

Regarding other dietary sources of iodine, we were able to confirm the valuable role of cows’ milk [15,16] while, surprisingly, the role of IS was found to be marginal, which we think is due to the scarce awareness of iodine prophylaxis campaigns that emerged from our questionnaire, as in other surveys in our region [16]. In our opinion, this lack of awareness—as low as 10% of the women interviewed—impacts on the effective regular use of iodised salt, aside from the declared use/non-use. However, it emerged from the multivariate analysis that only ICS use was an independent predictor of iodine sufficiency, although it is worth noting that it was not independently associated with iodine excess, suggesting that it is safe also in women with AT on L-T4 therapy.

As expected, urinary iodine excretion and ICS did not play a role in maternal thyroid function, since L-T4 use was found to be the only predictor for maternal TSH levels. Pregnancy outcomes and newborn anthropometric factors were not associated with iodine status, as expected in an area of mild-to-moderate iodine deficiency. Indeed, while there is a clear association between urinary iodine excretion and maternal and neonatal outcomes in cases of severe iodine deficiency, this association is debatable in areas of only mild-to-moderate deficiency. Consideration of many more factors and a larger sample are needed to explore this fine relationship [17,18,19], and this goes beyond the aims of this study.

Regarding neonatal TSH, it is interesting that, in agreement with Jastrzębska et al.’s study, we found higher neonatal TSH levels among the offspring of supplemented women than among those of non-supplemented women, although the levels were always within the normal reference range. Neonatal TSH is known to reflect the iodine supply during foetal life [9,10]. A TSH higher than 5 mUI/L in more than 3% of a sample is considered a proxy of iodine deficiency in that population [4]. However, in ours, as in other studies [12,14], a higher neonatal TSH seems to be a reflection of a higher iodine intake. A possible explanation involves an inhibitory effect of iodine on the foetal thyroid [20], a striking effect that can be more evident in areas of iodine deficiency, like ours [21]. These data offer further evidence that the goal of iodine prophylaxis campaigns should be the extensive use of IS, possibly along with daily dairy consumption, while supplements should ideally play a marginal role once iodine sufficiency in a population is reached.

In contrast to Jastrzębska et al.’s study, we decided to focus our study only on women with autoimmune thyroiditis, and not more generally on pregnant women undergoing L-T4 therapy, and to also include women not on L-T4 in order to investigate the effects of a compromised thyroid on urinary iodine clearance. A part of urinary iodine excretion is known to also be related to thyroid iodine clearance, the intensity of which is dependent on iodine-replete status [22]. We aimed to ascertain if, in autoimmune thyroiditis, the inflammatory and fibrotic process may lead to the impairment of iodine thyroid clearance, with a possible increase in the iodine eliminated with urine. Interestingly, but with the caveat of the low number of non-treated women with thyroiditis, all of whom were ICS-supplemented, the median UI/Creat was adequate—neither more than adequate nor excessive. However, being untreated, these are women with autoimmune thyroiditis but still functioning thyroids. The analysis of urinary iodine excretion in relation to the duration of the thyroiditis diagnosis and the size of the thyroid revealed a negative but non-significant correlation, suggesting that the impact of the autoimmune process on UI/Creat, if present, is at least marginal.

Our study has a series of limitations. The first concerns its retrospective design. Moreover, data were collected from the 20th to the 29th gestational week, i.e., across the second and third trimesters, which could lead to heterogeneity in the data. The most important limitation is related to sample size, which, in our opinion, reduced the statistical power of the study. In addition, there was no control group of women taking neither iodine nor L-T4, and very few of the women were taking only an ICS and were not on L-T4. The present study aimed to shed light on a somewhat neglected issue, but more powerful studies are needed to reach solid conclusions.

## 5. Conclusions

Even taking into account all the limitations of our study, we conclude that pregnant women residing in areas of mild-to-moderate iodine deficiency with AT and on L-T4 therapy still need an ICS to reach iodine sufficiency. However, since L-T4 is itself a source of iodine, the amount of iodine supplemented should ideally be based on the dose of L-T4 being taken.

## Figures and Tables

**Figure 1 nutrients-17-00542-f001:**
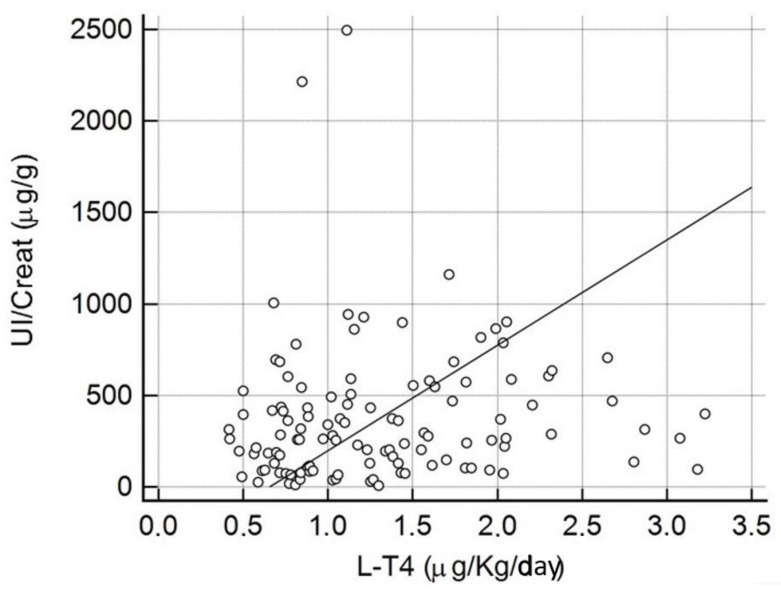
Correlation between UI/Creat and the daily L-T4 dosage (μg/Kg/day).

**Table 1 nutrients-17-00542-t001:** Subjects’ characteristics. IQR: interquartile range.

Characteristics		
Personal and anthropometric data		
Age (years): median, IQR		35.0, 31.6–38.3
Pregnancy data		
Gestational age at enrolment (weeks): median, IQR		24, 20.0–29.5
Knowledge of iodine prophylaxis campaign	Yes	12/122 (9.8%)
	No	110/122 (90.2%)
Therapy		
Levothyroxine therapy	Yes	115/125 (92%)
	No	10/125 (8%)
Supplements		
Iodine-containing supplement	Yes	112/125 (89.6%)
	No	13/125 (10.4%)
Dietary habits		
Use of iodised salt	Yes	94/122 (77.1%)
	No	28/122 (22.9%)
Cows’ milk, at least 1 cup/day	Yes	45/122 (36.9%)
	No	77/122 (63.1%)
Cheese, more than once a week	Yes	57/122 (46.7%)
	No	65/122 (53.3%)

**Table 2 nutrients-17-00542-t002:** Median UI/Creat and frequency of UI/Creat ≥ 150 μg/g according to social and dietary factors.

Characteristics		Median UI/Creat, μg/g (IQR)	*p*	UI/Creat ≥ 150, N (%)	*p*
Knowledge of iodine prophylaxis campaign	Yes	263.5 (151.5–476.00)	0.9 ^a^	9/12 (75.0)	1 ^f^
	No	266.00 (128.25–497.0)		78/110 (70.9)	
Levothyroxine therapy	Yes	267.0 (117.0–498.5)	0.6 ^b^	80/115 (69.6)	0.3 ^g^
	No	175.0 (164.5–378.1)		9/10 (90.0)	
Use of iodised salt	Yes	266.0 (150.3–519.5)	0.6 ^c^	70/94 (74.5)	0.2 ^h^
	No	248.5 (90.5–444.5)		17/28 (60.7)	
Iodine-containing supplements	Yes	271.5 (134.3–494.8)	0.1 ^d^	83/112 (74.1)	0.03 ^i^
	No	148.0 (86.0–281.0)		6/13 (46.2)	
Cows’ milk, at least 1 cup/day	Yes	342.0 (195.0–603.0)	0.03 *^e^	37/45 (82.2)	0.04 ^l^
	No	239.0 (115.0–436)		50/77 (64.9)	

^a, b, c,^ * Cohen’s d index for the effect size: 0.306 (95% CI: 0.675–0.065). *p* after false discovery rate test (FDR) with Benjamini–Hochberg method for multiple tests: a: 0.9; b: 0.75; c: 0.75; d: 0.25; e: 0.15; f: 1; g: 0.38; h: 0.33: i: 0.1; l: 0.1.

**Table 3 nutrients-17-00542-t003:** More than adequate UI/Creat levels and associations with the main urinary iodine determinants.

UI/Creat ≥ 250 μg/g, N = 69			N (%)	*p*
	Levothyroxine therapy	Yes	65/69 (94.2)	0.3
		No	4/69 (5.8)	
	Use of iodised salt	Yes	54/68 (79.4)	0.5
		No	14/68 (20.6)	
	Iodine-containing supplements	Yes	65/69 (94.2)	0.06
		No	4/69 (5.8)	
	Cows’ milk, at least 1 cup/day	Yes	30/68 (44.1)	0.06
		No	38/68 (55.9)	

**Table 4 nutrients-17-00542-t004:** Excessive UI/Creat levels and associations with the main urinary iodine determinants.

UI/Creat ≥ 500 μg/g, N = 31			N (%)	*p*
	Levothyroxine therapy	Yes	30/31 (96.8)	0.3
		No	1/31 (3.2)	
	Use of iodised salt	Yes	26/31 (83.9)	0.3
		No	5/31 (16.1)	
	Iodine-containing supplements	Yes	29/31 (93.5)	0.4
		No	2/31 (6.5)	
	Cows’ milk, at least 1 cup/day	Yes	16/31 (51.6)	0.05
		No	15/31 (48.4)	

## Data Availability

The data supporting the findings of this study are available on request from the corresponding author, Caterina Mian.

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
