# Peer review of "Women with Autoimmune Thyroiditis Taking Levothyroxine During Pregnancy: Is Iodine Supplementation Needed?"

_nutrients, 2025, doi:10.3390/nu17030542_

Round 1
Reviewer 1 Report
Comments and Suggestions for Authors
The concept of Censi et al is interesting. However, the study has important weaknesses that need to be addressed.
Major observations
--- The group of women taking iodine supplements is extremely small compared to the group taking both thyroxine and iodine supplements, despite the statistical findings
--We need a control group that takes neither thyroxine nor supplements, at least to have real data on the daily intake of iodine in the region.
-- I'd rather eliminate the data on ethnicity, education, smoking, dietary habits, because they don't contribute to the main objective of the study, but they make the M.S. "heavy".
Minor observation
--In the Discussion Section and elsewhere some data are not supported by literature such as lines 330-331, 343-345
--No bibliographic data from the web are accepted.
--- The English syntax needs to be improved
Comments on the Quality of English Language
The English syntax needs to be improved
Author Response
The concept of Censi et al is interesting. However, the study has important weaknesses that need to be addressed.
Major observations
--- The group of women taking iodine supplements is extremely small compared to the group taking both thyroxine and iodine supplements, despite the statistical findings
- We agree with the Reviewer and it represents for sure a major limitation of the present study. Our study aimed to shed a light on the pretty neglected issue of iodine supplementation in pregnancy in women with thyroiditis already taking iodine, but more powerful studies are needed to get to solid conclusions. Now this limitation of the study has been stressed in the Discussion section, lines 270-274.
--We need a control group that takes neither thyroxine nor supplements, at least to have real data on the daily intake of iodine in the region.
- We agree with the Reviewer and it represents for sure a major limitation of the present study. Likewise in point 1 of the reviewer observation, this issue has been stressed in the Discussion section, lines 270-274.
-- I'd rather eliminate the data on ethnicity, education, smoking, dietary habits, because they don't contribute to the main objective of the study, but they make the M.S. "heavy".
We welcome Reviewer’s suggestion and we deleted this data accordingly. In particular, data regarding BMI, ethnicity, education and smoking were deleted. As regards dietary habits, only data regarding cow’s milk was kept, since it revealed interesting at univariate analysis and on the basis of the known importance of cow’s milk on the determination of iodine status. As a consequence of the elimination of cheese from determinants, also multivariable analysis of logistic regressions was re-evaluated, as shown in results section.
Minor observation
--In the Discussion Section and elsewhere some data are not supported by literature such as lines 330-331, 343-345
We apologize and bibliographic references were now added wherever appropriate. Regarding 343-345, the amount of excreted iodine in case of autoimmune thyroiditis was a question we wanted to address rather than an information already present in Literature, the sentence has been changed to be more clear (lines 261-263).
--No bibliographic data from the web are accepted.
We apologize and bibliographic references were now corrected, with the exception of the guidelines of the WHO, that, at the best of our knowledge, are available only as online PDF.
--- The English syntax needs to be improved
The manuscript has now been revised by a mother-tongue English speaker.

Reviewer 2 Report
Comments and Suggestions for Authors
The topic of the research is important, and the treatment of non-normally distributed data seems to have been conducted reasonably.
The lack of large-sample randomization creates a number of difficulties that are addressed below. It is not clear if those issues are fixable.
The p-values in general are not impressive, as they are just on the fringe of the usual 0.05 Type I error criterion for statistical significance. A more functional approach would be to report effect size metrics such as Cohen's d.
The statement in lines 32 and 33 about "a positive correlation at the borderline of statistical significance" is pretty much an oxymoron. By the usual 0.05 p-value criterion the null hypothesis of zero correlation cannot be rejected; hence the correlation can't really be said to have a (positive) direction. This point is illustrative of a major problem with the results in that pretty much every statement of association is very shaky given the absence of definitive evidence.
The last line of Table 1 has a typo: "437122 (53.3)" apparently should be "65/122 (53.3)". Please check all reported calculations.
Table 1 also produces another major issue with data interpretation. Although a reported p value of 0.03 would qualify by itself as being statistically significant, the table shows a large number of test results. Repeated testing carries the near-certainty that some results will be false positives. Applying the Bonferroni correction to adjust for inflated Type I error may well show that none of the reported results are anywhere close to statistical significance. An alternative is to apply the false discovery rate adjustment, which well may lead to the same outcome. Please check all the reported multiple-testing p-values for this point.
It would be ideal for the manuscript to include a robust discussion of implications of the results for health policy and clinical practice. Doing so becomes particularly difficult, though, with the paucity of statistically meaningful findings. Effect size metrics may help in that respect, but it seems likely that it would be necessary to talk somewhat awkwardly about implications associated with null findings; "the dog that didn't bark" from Sherlock Holmes fame could be an approach.
Author Response
The topic of the research is important, and the treatment of non-normally distributed data seems to have been conducted reasonably.
The lack of large-sample randomization creates a number of difficulties that are addressed below. It is not clear if those issues are fixable.
We agree with the Reviewer and it represents for sure a major limitation of the present study. Our study aimed to shed a light on the pretty neglected issue of iodine supplementation in pregnancy in women with thyroiditis already taking iodine, but more powerful studies are needed to get to solid conclusions. Now this limitation of the study has been stressed in the Discussion section, lines 270-274.
The p-values in general are not impressive, as they are just on the fringe of the usual 0.05 Type I error criterion for statistical significance. A more functional approach would be to report effect size metrics such as Cohen's d.
We have now reported the Cohen’s d index for effect size in the case of comparison between continuous variables resulting in a p <0.05 (applying to Cows’ milk and UI/Creat), line 159 and table 2.
The statement in lines 32 and 33 about "a positive correlation at the borderline of statistical significance" is pretty much an oxymoron. By the usual 0.05 p-value criterion the null hypothesis of zero correlation cannot be rejected; hence the correlation can't really be said to have a (positive) direction. This point is illustrative of a major problem with the results in that pretty much every statement of association is very shaky given the absence of definitive evidence.
The reviewer is right, it cannot be stated. The sentence has been modified, in the abstract (lines 23-24) and in Results section accordingly (lines 138-139).
The last line of Table 1 has a typo: "437122 (53.3)" apparently should be "65/122 (53.3)". Please check all reported calculations.
We apologize and thank the reviewer. It has been fixed.
Table 1 also produces another major issue with data interpretation. Although a reported p value of 0.03 would qualify by itself as being statistically significant, the table shows a large number of test results. Repeated testing carries the near-certainty that some results will be false positives. Applying the Bonferroni correction to adjust for inflated Type I error may well show that none of the reported results are anywhere close to statistical significance. An alternative is to apply the false discovery rate adjustment, which well may lead to the same outcome. Please check all the reported multiple-testing p-values for this point.
According to the Reviewer's suggestion, we applied the false discovery rate adjustment and we reported in Table 2 the p adjusted for Benjamini–Hochberg method, corrected for multiple tests. After the manuscript revision and according to Reviewer 1, several items have been deleted from the study. Please note that only 5 items were finally subjected to our interest, and are all clinically related to the main issue of the study and clinically-driven. As a consequence, given the clinical substance at the basis of these statistics, in the Results and Discussion sections we kept referring to the uncorrected p, although reporting the adjusted p in the Table and including the issue of multiple comparison as potential limitation that could be addressed by future replication studies. Indeed, it is important to consider that iodine status, iodine prophylaxis campaign knowledge, L-T4 therapy, iodized salt use, iodine-containing supplements and cows’ milk use are all clinically relevant features, in which a potential role is expected based on the available literature on the issue. We hope that the reviewer will find our choice to be reasonable.
It would be ideal for the manuscript to include a robust discussion of implications of the results for health policy and clinical practice. Doing so becomes particularly difficult, though, with the paucity of statistically meaningful findings. Effect size metrics may help in that respect, but it seems likely that it would be necessary to talk somewhat awkwardly about implications associated with null findings; "the dog that didn't bark" from Sherlock Holmes fame could be an approach.
The Reviewer is right about the paucity of solid results coming from our study and thus possible conclusions for health policies. Our study does not offer definitive conclusions, but wants to bring clinicians’ attention to an issue that is pretty neglected in Health policies: does the AT patients in L-T4 treatment need iodine supplementation during pregnancy and how? The limits of our study have been now stressed more clearly in Discussion section.

Round 2
Reviewer 2 Report
Comments and Suggestions for Authors
The authors have been responsive to issues addressed by the reviewers.